# Synthesis of Estrone Heterodimers and Evaluation of Their In Vitro Antiproliferative Activity

**DOI:** 10.3390/ijms25084274

**Published:** 2024-04-12

**Authors:** Noémi Bózsity, Viktória Nagy, Johanna Szabó, Balázs Pálházi, Zoltán Kele, Vivien Resch, Gábor Paragi, István Zupkó, Renáta Minorics, Erzsébet Mernyák

**Affiliations:** 1Institute of Pharmacodynamics and Biopharmacy, University of Szeged, Eötvös u. 6, H-6720 Szeged, Hungary; bozsity-farago.noemi@szte.hu (N.B.); nagy.viktoria07@gmail.com (V.N.);; 2Department of Analytical and Molecular Chemistry, University of Szeged, Dóm tér 8, H-6720 Szeged, Hungary; johanna.szab@gmail.com (J.S.); balazs.palhazi@gmail.com (B.P.); 3Department of Medicinal Chemistry, University of Szeged, Dóm tér 8, H-6720 Szeged, Hungary; kele.zoltan@med.u-szeged.hu (Z.K.); resch.vivien.erzsebet@szte.hu (V.R.); paragi@gamma.ttk.pte.hu (G.P.); 4Institute of Physics, University of Pécs, Ifjúság útja 6, H-7625 Pécs, Hungary; 5Department of Theoretical Physics, University of Szeged, Tisza Lajos krt. 84-86, H-6720 Szeged, Hungary; 6Institute of Pharmacognosy, University of Szeged, Eötvös u. 6, H-6720 Szeged, Hungary

**Keywords:** 13α-estrone, D-secoestrone, heterodimer, antiproliferative effect, tubulin polymerization, taxoid binding site of tubulin

## Abstract

Directed structural modifications of natural products offer excellent opportunities to develop selectively acting drug candidates. Natural product hybrids represent a particular compound group. The components of hybrids constructed from different molecular entities may result in synergic action with diminished side effects. Steroidal homo- or heterodimers deserve special attention owing to their potentially high anticancer effect. Inspired by our recently described antiproliferative core-modified estrone derivatives, here, we combined them into heterodimers via Cu(I)-catalyzed azide–alkyne cycloaddition reactions. The two *trans*-16-azido-3-(*O*-benzyl)-17-hydroxy-13α-estrone derivatives were reacted with 3-*O*-propargyl-D-secoestrone alcohol or oxime. The antiproliferative activities of the four newly synthesized dimers were evaluated against a panel of human adherent gynecological cancer cell lines (cervical: Hela, SiHa, C33A; breast: MCF-7, T47D, MDA-MB-231, MDA-MB-361; ovarian: A2780). One heterodimer (**12**) exerted substantial antiproliferative activity against all investigated cell lines in the submicromolar or low micromolar range. A pronounced proapoptotic effect was observed by fluorescent double staining and flow cytometry on three cervical cell lines. Additionally, cell cycle blockade in the G2/M phase was detected, which might be a consequence of the effect of the dimer on tubulin polymerization. Computational calculations on the taxoid binding site of tubulin revealed potential binding of both steroidal building blocks, mainly with hydrophobic interactions and water bridges.

## 1. Introduction

The development of drugs acting in a selective manner is one of the major challenges in modern medicinal chemistry. This is the case in cancer treatment as well, where drugs with poor tolerability profiles cannot be excluded from the current therapeutic guidelines. Structural modifications of naturally occurring biologically active substances may result in novel, selective therapeutic agents. Natural product hybrids are often more effective and selective than the parent compounds [1]. The components of drug candidates constructed from diverse molecular entities may result in synergic action with diminished side effects. Steroid dimers represent an exciting class of compounds with unique physicochemical profiles, which enables their application in diverse fields [2]. Besides liquid crystals and detergents, cytotoxic steroid dimers are also known [3,4,5,6]. Cephalostatins and ritterazines as natural product hybrids consist of a pyrazine unit and highly oxygenated steroid moieties [7,8]. Their high anticancer activity is entirely different from those of their separate units. *bis*-Estradiol derivative E2D connected via a *bis*-triazolylpyridine unit (Figure 1) possesses unique cytotoxic activities on certain cancer cell lines with antimitotic properties [9]. Estradiol dimer E2D inhibits tubulin assembly in a low micromolar range, thus having an antimitotic activity more robust than that of the 2-methoxyestradiol derivative. Computational simulations revealed the binding of the estradiol dimer to the colchicine binding site in the tubulin dimer. It was shown by Drasar et al. that both the nature and the size of the linker between the two estradiol parts substantially modify cytotoxicity and the cell cycle profiling of the compounds [10].

Antitubulin agents represent a highly featured class of anticancer chemotherapeutics [11]. Two types of compounds belong to this group: microtubule stabilizing (MSAs) and destabilizing agents (MDAs) [12,13]. Most MDAs usually bind to the colchicine binding site of the tubulin dimer; however, MSAs bind to the taxoid binding site [14]. The triazole ring is one of the common structural elements in antitubulin compounds [15].

Core-modified estrone derivatives represent a class of biologically active semisynthetic compounds with suppressed estrogenic behavior [16,17,18,19]. Opening the D-ring or configuration inversion of C-13 on the estrane core results in a conformational change, which does not allow the compounds to bind effectively to the nuclear estrogen receptors. Therefore, this family of steroidal agents may allow the development of compounds with selective biological activity other than estrogenic. We recently published the synthesis and in vitro pharmacological investigation of D-secoalcohol estrone triazole **3** (Figure 2) as a promising anticancer agent [20]. We described its low micromolar-cell-growth-inhibitory potential against human adherent cancer cell lines of gynecological origin. The in vitro mechanistic investigations revealed its microtubule-stabilizing activity. Core-modified estrone alkynes and azides were synthesized and transformed via Cu(I)-catalyzed azide-alkyne click reaction (CuAAC) as a key step to explore the influence of the structure modification on the antitumoral effect. Our comparative pharmacological investigations suggest that a 3-*O*-benzyl moiety on the D-secoestrone alcohol or oxime (in compound **1** or **2**, Figure 2) is advantageous concerning the antiproliferative potential [21,22]. However, this effect increased by introducing a triazole ring between the steroid and the benzyl group (compound **3** or **4**, Figure 2) [20,22]. The moderate antiproliferative activity of *trans*-azidoalcohols (**5** and **6**) developed recently could be highly improved by click conjugations with (*subst.*)phenylacetylene reagents [23]. An important structure–activity correlation appeared concerning the stereochemistry of 16-triazoles **7** and **8**. The 16β,17α counterparts exerted substantial anticancer potential exclusively.

Based on these promising results, here, we report the development of potential anticancer candidates by transforming the most active monomers (see Figure 2) into heterodimers. The two *trans*-azidoalcohols (**5** or **6**) were intended to click with the propargylated derivative of D-secoalcohol **1** or oxime **2** to obtain valuable structure–activity relationships. Our next goal was to investigate the in vitro cell-growth-inhibitory potential of the newly synthesized heterodimers against human adherent cancer cell lines (Hela, SiHa, C33A, MCF-7, T47D, MDA-MB-231, MDA-MB-361, A2780). In addition, exploring the mechanism of action of the most potent heterodimer was also planned. Computational simulations were also included to describe the potential binding interactions of the most potent dimer with the taxoid binding site of tubulin.

## 2. Results

### 2.1. Synthesis of the Heterodimers *(**11**–**14**)*

Azide (**5** and **6**) and alkyne (**9** and **10**) counterparts of the appropriate core-modified estrone derivatives were synthesized according to recently elaborated procedures [21,23,24]. D-Seco compounds were transformed into alkynes (**9** or **10**) via the propargylation of their phenolic hydroxy function [21,22]. Syntheses of the *trans*-azidoalcohols (**5** or **6**) were achieved in a two-step procedure starting from the Δ^16^ derivative of 13α-estrone 3-benzyl ether [23]. The CuAAC reactions of the alkynes (**9** or **10**) and azides (**5** or **6**) were carried out utilizing the coupling method published earlier [22], using CuI as a catalyst, PPh_3_ as an accelerating ligand, and DIPEA as a base in toluene as solvent (Figure 1). The heterodimers were obtained in high yields after column chromatography (87–94%). All four dimers could efficiently be recrystallized from methanol as a solvent. The structure elucidation of dimers **11**–**14** was implemented by NMR spectroscopy.

### 2.2. Pharmacology

The results of the antiproliferative MTT assay revealed that only a single compound (**12**) of the four investigated heterodimers (**11**–**14**) was active against the used cancerous cell lines. Compound **12** had a substantial growth-inhibitory effect on all investigated cell lines, with inhibition values greater than 80% at 30 μM. Compounds **11** and **13** did not exhibit any significant antitumor effect, even at the higher concentration. They were unable to reach even the 50% inhibition value and, in most cases, their effect was lower than 30%. In the case of **14**, there was slight inhibition at 30 µM on SiHa, MCF-7, T47-D, and MDA-MB-231 cell lines (Table 1).

Based on its high antiproliferative potency, **12** was selected for detailed investigations. Sigmoidal dose–response curves were fitted, and the IC_50_ values were found to be lower than 3.5 µM on all tested cell lines (Table 1).

The investigation of tumor selectivity in intact human fibroblast (MRC-5) and intact mouse fibroblast (NIH/3T3) cell lines revealed that **12** has an antiproliferative effect similar to that of cisplatin [24], a drug routinely used in tumor therapy.

The tumor selectivity index results revealed that in the case of the non-malignant NIH/3T3 cell line, the selectivity of our compound **12** is better on all investigated tumor cell lines compared with the clinically used cisplatin. In the case of non-malignant MRC-5 cells, the results show a selectivity comparable with that of cisplatin and with a similar range of selectivity indices (comp **12**: 0.36–1.28 and cispl: 0.24–3.47). In the case of HeLa, SiHa, T47D, and MDA-MB231, comp **12** shows better selectivity, while in the case of C33A, A2780, and MDA-MB-361, cisplatin shows better selectivity. Furthermore, MCF-7 cells have almost the same selectivity indices (Table 2).

Based on our previous studies of the D-secoestrone building block [20], we chose the cervical panel (Hela, SiHa, and C33A cells) for further investigations. The morphological changes in the three cervical cancer cell lines induced by the investigated heterodimer **12** are presented in Figure 3. The presence of the living, early apoptotic, and late apoptotic (secondary necrotic) cells was distinguished according to cell morphology and membrane integrity. Two separate pictures from the same field were taken to detect the form of cell death.

After 24 h of incubation, a significant elevation in the early apoptotic cell ratio was detected in HeLa and C33A cells. In contrast, in the case of SiHa cells, the late apoptotic population increased, which was only visible after treatment at high concentrations.

Flow cytometric analyses were performed for the quantitative characterization of the cell cycle changes caused by **12**. The results on the three cervical cell lines with different HPV statuses show a similar trend (Figure 4). After 24 h of incubation, **12** elicited slight changes in the ratios of cells in the different cell cycle phases, especially in the case of HeLa and C33A cells. In turn, the analysis shows statistically significant alterations in the cell cycle distribution on the SiHa cell line; that is, an increase at the G1 and a decrease at the S phase can be observed. Longer incubation resulted in substantial changes in the cell cycle distribution on all three investigated cell lines. Regardless of the HPV status of the cell lines, the elevation of the apoptotic subG1 phase was observed; moreover, a significant and dose-dependent increase was detected in the number of cells in the G2/M phase, followed by a G1 phase reduction.

The direct effect of our test compound **12** on tubulin polymerization is presented in Figure 5. The kinetic curve indicates that **12** accelerates microtubule formation in the growth phase, which is also reflected in the higher calculated V_max_ values. For the statistical analysis, the maximum rate of microtubule formation was calculated, where V_max_ is the highest difference in the absorbance of two successive time points at the kinetic curve (Δabsorbance/min). Our results show a statistically significant elevation in the Vmax value of test compound **12** compared to that of the negative control tubulin. This is similar to the positive control paclitaxel, a well-known microtubule stabilizer.

### 2.3. Computational Simulations

The separated ligand conformations of heterodimer **12** in water and DMSO solutions were investigated. The two steroid building blocks did not interact with each other in the solvents investigated. We found that when fitting the ligand into the central triazole ring, it misses some torsional angle values in DMSO compared to the water-dissolved conformational ensemble (Appendix A). The docking results of compound **12** to the taxoid binding site of tubulin show that the dimer might bind to this site with both of its building blocks. We also modeled the binding of the D-secoestrone part in the binding pocket. The dG values were found to be –54.92 ± 3.34 and –58.27 ± 4.18 kcal/mol, according to the MM/GBSA calculations, taking the open and the closed D-ring bound starting positions, respectively. In the binding pocket, the ligand showed more compact geometry, since the two building blocks were more often close to each other than in bulk water or DMSO solution. Protein–ligand interactions were also investigated using the Simulation Interaction Diagram analysis tool of the Schrodinger Desmond package. Mainly hydrophobic interactions and water bridges were found to determine the ligand position (Appendix A). In the case when the starting pose was provided by the docking calculation, the REST simulation revealed that the most important interactions were formed with Phe-272, Arg-369, and Tyr-283 amino acids (Figure 6).

Taking the D-secoestrone part-bound starting geometry, we found a more modest connection between the ligand and the residues. The characteristic interactions (Ala-233, Arg-320, Appendix A) were formed with smaller fractions along the trajectory.

## 3. Discussion

Core-modified D-seco- and 13α-estrone compound families have been the main focus of our recent research on anticancer agents. The steroid derivatives have been modified with alkyne or azide moieties to obtain suitable starting compounds for CuAAC reactions [20,21,22,23,24]. The introduction of a terminal alkyne function was achieved via the propargylation of the phenolic hydroxy function. Azides were synthesized via epoxide ring-opening reactions at ring D, utilizing sodium azide as a reagent. We synthesized a compound library for structure–activity relationship studies. The steroidal alkynes or azides were reacted with smaller azide or alkyne coupling partners. The CuAAC reactions of the steroidal alkynes with (*subst.*)benzyl azides afforded triazoles exhibiting low nanomolar or submicromolar cell-growth-inhibitory potentials against human adherent cancer cell lines of gynecological origin [20,21,22]. This biological activity significantly depended on the nature and position at C-13 of the functional group. It was found that 13β-D-Secoestrone derivatives proved to be more potent than their 13α-counterparts. Concerning the 16-azido compounds (**5**, **6**), their coupling to (*subst.*)phenylacetylenes seemed to be advantageous, with the 16β,17α-isomers (**8**) being more active [23]. With these structure–activity relationships in mind, we intended to conjugate the structural moieties identified earlier into heterodimeric compounds. Four dimers (**11**–**14**) were synthesized, differing in positions of the 16-triazolyl and 17-hydroxy moieties and in the nature of the functional group at C-13 of the D-seco counterpart. The CuAAC methodology elaborated earlier was suitable for the reactions of the steroidal coupling partners without any modification [22]. The NMR investigation of the heterodimers (**11**–**14**) was based on the spectra of the monomers (**5**, **6**, **9**, **10**). The chemical transformations of the starting compounds via CuAAC reactions did not influence the other parts of the reactants, including the chiral carbon atoms. The ^1^H and ^13^C NMR spectra of the dimers (**11**–**14**) reflect the signals specific to the corresponding molecular entities well (Appendix A).

Considering the antiproliferative activity of the corresponding monomers, we investigated the growth-inhibitory effect of the four newly synthesized dimer derivatives **11**–**14**. Our growth-inhibitory results revealed that the 16β,17α positions on the azide building block are required for any antiproliferative effect. Additionally, the hydroxymethyl group at the C-13 of the D-seco counterpart is advantageous over the oxime moiety. The investigated **12** dimer displayed a potent cell-growth-inhibitory effect on all cell lines, including the cervical cells, independently of their HPV status. On the contrary, the recently reported D-secoestrone triazole, which might be considered to be a monomer of the active heterodimer **12**, inhibited the proliferation of HeLa and C33A cells, but was not able to inhibit the growth of the HPV 16+ SiHa cells. The tumor selectivities of monomers and heterodimer **12** were comparable to that of the clinically used cisplatin, as determined on non-cancerous fibroblasts (MRC-5 and NIH/3T3). Cervical cells were selected for the detailed pharmacological studies based on the similarity in chemical structure and pharmacological effect of dimer **12** against gynecological cancer cells and D-secoalcohol **3**.

At the beginning of the apoptotic process, chromatin condensation and nuclear fragmentation occur without membrane damage. However, under in vitro conditions, the elimination of damaged cells is insufficient, because of the absence of phagocytic capacity; therefore, the cells lose the integrity of their membrane [25]. Our findings revealed morphological changes that confirm the concentration-dependent proapoptotic effect of **12** on the investigated cell lines. Briefly, **12** induced chromatin condensation on HeLa and C33A cells. In the case of SiHa cells, the elevation of the concentration led to the formation of late apoptotic cells with loss of the membrane function.

The cell cycle analysis results indicated that **12** causes a blockade in the G2/M phase of the cell cycle, and the morphological changes mentioned previously were also confirmed. Therefore, progressive DNA content elimination was observed. DNA fragmentation becomes evident after 48 h with the rising subG1 population, while it is also visible in the G1 elevation after 24 h. Our hypothesis is that if the cells enter into the apoptotic mechanism from the G2/M phase, the DNA fragmentation starts from a double DNA content, resulting in particles close to the single DNA content and can mimic cells in the G1 or S phase. As the process moves forward, subG1-sized particles appear, which require a longer time, and this may also be the reason for the delayed subG1 elevation. A similar effect can also be found in the literature with other estrogen-related compounds, such as some A-ring-modified sulphamoylated analogs or the D-ring-modified D-homoestrone [26].

However, previous studies described G2/M arrest as a result of different estrogen-derivative treatments. Interestingly, however, their primary mechanism of action is different. It has been reported that the D-ring-expanded D-homoestrone-induced G2/M accumulation is related to G2-phase blockade due to checkpoint failure. At the same time, the widely investigated 2-ME evokes abnormal spindle formation during the M phase, therefore inhibiting tubulin polymerization due to its interaction with the colchicine binding site of β-tubulin [27,28]. Moreover, our former findings revealed that D-secoestrone-triazole **3**, which is a part of our present pharmacologically active heterodimer **12**, also induces G2/M blockade in the M phase. However, it has a mechanism different than those described earlier. D-secoestrone-triazole enhanced the formation of microtubules from tubulin dimers [20].

Considering that the background of the G2/M arrest provoked by **12** is similar to its monomer component [20], an in vitro tubulin polymerization assay was performed. Kinetic measurements proved our hypothesis. Our experimental data confirm that the M-phase blockade caused by **12** is closely related to its direct accelerating effect on tubulin polymerization. The imbalance between the polymerization and depolymerization of microtubules leads to fatal consequences of the termination of the cell division [29]. Compound **12** shifts the dynamic equilibrium towards microtubule stabilization, leading to aberrant mitosis and causing a blockade in cell cycle progression, triggering the beginning of the apoptotic process.

Computational simulations were performed to gain insight into the binding characteristics of the most potent heterodimer **12** into the taxoid binding site of tubulin protein. Compound **12** consists of two relatively flexible steroidal building blocks, and the methylene bridge between them provides an additional high degree of flexibility. Considering these structural features, the separated ligand conformations were investigated in two solvents used in the biological assays. Although we found some differences in the torsional angle values calculated in DMSO or water, it can be stated that the two steroidal building blocks do not tend to interact with each other in a pure solvent environment. Based on this result, we modeled the binding of both steroidal parts of dimer **12**. The MM/GBSA calculations confirmed that the dG energy difference between the two ligands posed is less significant than the simple docking results suggested. Furthermore, the findings also draw attention to the flexibility of the taxoid binding site, as it is formed mainly by multiple flexible loops. The predominance of hydrophobic interactions and water bridges was observed while investigating meaningful protein–ligand interactions. Interestingly, although the ligand supports several hydrogen-bonding opportunities, none acted as a significant interaction in forming the protein–ligand complex. Taking the opened D-ring-bound starting geometry, we found a more modest connection between the ligand and the residues. It seems that according to REST simulations, dimer **12** cannot provide a unique binding position. Both building blocks can bind to the binding pocket and they remain stable, and the geometry of the bound ligand is less elongated compared to the solvated unbound situation.

## 4. Methods and Materials

### 4.1. Chemistry

Melting points (Mps) were determined with a Kofler hot-stage apparatus and were uncorrected. Elemental analyses were performed with a Perkin-Elmer CHN analyzer model. Thin-layer chromatography was performed on silica gel 60 F254 (layer thickness 0.2 mm, Merck, Rahway, NJ, USA); eluents: (A) 70% ethyl acetate/30% hexane, (B) 30% ethyl acetate/70% hexane. The spots were detected with I_2_ or UV (365 nm) after spraying with 5% phosphomolybdic acid in 50% aqueous phosphoric acid and heating at 100–120 °C for 10 min. Flash chromatography was performed on silica gel 60, 40–63 μm (Merck). ^1^H NMR spectra were recorded in DMSO-d_6_ or CDCl_3_ solution with a Bruker DRX-500 instrument at 500 MHz. ^13^C NMR spectra were recorded with the same instrument at 125 MHz under the same conditions. Mass spectrometry: Full-scan mass spectra of the newly synthesized compounds were acquired in the range of 100 to 1100 m/z with a Q Exactive Plus quadrupole-orbitrap mass spectrometer (Thermo Fisher Scientific, Waltham, MA, USA) equipped with a heated electrospray (HESI). Analyses were performed in positive ion mode using flow-injection mass spectrometry with a mobile phase of 50% aqueous acetonitrile containing 0.1 v/v% formic acid (0.3 mL/min flow rate). Aliquots of 5 µL of samples were injected into the flow. The ESI capillary was adjusted to 3.5 kV, and N_2_ was used as a nebulizer gas.

#### General Procedure for the Synthesis of Heterodimers (**11**–**14**)

To a stirred solution of 16α-azido-3-benzyloxy-13α-estra-1,3,5(10)-trien-17β-ol **5** (100 mg, 0.25 mmol) or 16β-azido-3-benzyloxy-13α-estra-1,3,5(10)-trien-17α-ol **6** (100 mg, 0.25 mmol) in toluene (5 mL), Ph_3_P (13 mg, 0.05 mmol), CuI (4.7 mg, 0.025 mmol), DIPEA (0.13 mL, 0.75 mmol), and the appropriate terminal alkyne **9** or **10** (1 equiv.) were added. The reaction mixture was treated under reflux conditions for 2 h, allowed to cool, and evaporated in vacuo. The residue was purified by flash chromatography with EtOAc/hexane = 70/30 (for compounds **13** and **14**) or 30/70 (for compounds **11** and **12**) as eluent.

The characterization data for the reported compounds and NMR spectra are described in the Appendix A.

### 4.2. Pharmacology

#### 4.2.1. Antiproliferative (MTT) Assay

The growth-inhibitory effects of the test compounds were determined on 8 cancerous and 2 non-malignant cell lines. Human breast cancer cell lines (MCF-7, MDA-MB-231, MDA-MB-361, and T47D), ovarian carcinoma (A2780), HPV 18+ cervical adenocarcinoma (HeLa), and non-cancerous human and mouse fibroblast (MRC-5 and NIH/3T3) cell lines were purchased from ECACC (European Collection of Cell Cultures, Salisbury, UK), while SiHa (HPV 16+ squamous cell carcinoma) and C33 A (HPV—epithelial carcinoma) were purchased from ATCC (American Tissue Culture Collection, LGC Standards GmbH, Wesel, Germany). Cells were maintained in Eagle’s Minimum Essential Medium (EMEM) supplemented with 10% heat-inactivated fetal calf serum (FCS), 1% non-essential amino acids (NEAA), and 1% antibiotic-antimycotic mixture (AAM, penicillin–streptomycin). All media and supplements were obtained from Lonza Group Ltd. (Basel, Switzerland). The cells were maintained at 37 °C in a humidified atmosphere containing 5% CO_2_.

All cell types were seeded into 96-well plates at a density of 5000 cells/well and 10,000 cells/well in the case of C33A and MDA-MB-361. After overnight standing, increasing concentrations (0.1–30 µM) of the test compounds were added into the wells and incubated for 72 h at 37 °C under cell culturing conditions. Then, the cells were treated with 5.0 mg/mL MTT [3-(4,5-dimethylthiazol-2-yl)-2,5-diphenyltetrazolium bromide] solution for 4 h, the precipitated formazan crystals were dissolved in dimethyl sulfoxide, and the absorbance was read at 545 nm with a microplate reader; wells with untreated cells were utilized as controls [10]. Sigmoidal concentration–response curves were fitted to the measured points, and the IC50 values were calculated using GraphPad Prism 5.01 (GraphPad Software, San Diego, CA, USA). Cisplatin was used as a positive control in the same concentration range as the test compounds. Compound **12** was subjected to the MTT assay with intact, non-cancerous, human embryonal lung and mouse fibroblast (MRC-5 and NIH/3T3) cells under the same experimental conditions to determine the selective cytotoxic effect. Additionally, tumor-selectivity indices were determined by dividing the mean IC_50_ against non-malignant cells by the mean IC_50_ against tumor cells compared to both NIH/3T3 and MRC-5 cells.

#### 4.2.2. Cell Cycle Analysis by Flow Cytometry

Flow cytometric analysis was performed to determine the cellular DNA content of the treated cells. HeLa, C33A, and SiHa cells were seeded into 6-well plates at a density of 250,000–400,000 cells/well. After 24 or 48 h incubation, cells were harvested with trypsin and centrifuged at 1500 rpm for 10 min. The cells were fixed in 1 mL ice-cold 70% ethanol for 30 min and were stained with 0.1 mg/mL propidium iodide (PI) dye solution containing 0.02 mg/mL RNAse A for 60 min in the dark at room temperature. The cells were analyzed by a Partec CyFlow instrument (Partec GmbH, Münster, Germany). In each analysis, 20,000 events were recorded, and the percentages of the cells in the different cell cycle phases (subG1, G1, S, and G2/M) were determined using ModFit LT 3.3.11 software (Verity Software House, Topsham, ME, USA). The subG1 fraction was regarded as the apoptotic cell population [30].

#### 4.2.3. Hoechst 33258—Propidium Iodide (HOPI) Double Staining

HOPI staining was performed as described earlier [31]. The cervical cell lines (HeLa, SiHa, and C33A) were seeded in 96-well plates at 3000–5000 cells/well. After 24 and 48 h incubation with the test compounds, Hoechst 33258 (HO) and PI were added directly to the cells at final concentrations of 5 μg/mL and 3 μg/mL in EMEM, respectively. After 60 min incubation at 37 °C and 5% CO_2_, the cells were examined under a Nikon ECLIPSE TS100 fluorescence microscope (Nikon Instruments Europe B.V., Amstelveen, The Netherlands) with appropriate filters for HO (excitation: 360/40 nm bandpass filter, emission: 460/50 nm bandpass filter and 400 nm dichromatic mirror) and PI (excitation: 500/20 nm bandpass filter, emission: 520 nm long pass filter and 515 nm dichromatic mirror).

#### 4.2.4. Tubulin Polymerization Assay

The cell-independent direct effect of the test compound on tubulin polymerization was tested in vitro using a commercially available Tubulin Polymerization Assay Kit (Cytoskeleton Inc., Denver, CO, USA), according to the manufacturer’s recommendation. Briefly, 10 μL of the test compound was placed on a prewarmed UV transparent microplate. Then, 100 μL 3.0 mg/mL tubulin in 80 mM PIPES pH 6.9, 2 mM MgCl_2_, 0.5 mM EGTA, 1 mM GTP, and 10.2% glycerol was added to each sample, and the measurement was started immediately in a prewarmed UV spectrophotometer (SpectroStarNano, BMG Labtech, Ortenberg, Germany). A 50 min kinetic reaction was recorded to determine the absorbance of tubulin solution at 340 nm per minute. The maximum rate (Vmax, Δabsorbance/min) was calculated from the kinetic curve for the statistical analysis. Paclitaxel and general tubulin buffer were used as positive and negative controls.

### 4.3. Computational Simulations

The microtubule structure of 5SYF was downloaded from the protein database [32]. The crystal structure was prepared using the protein preparation wizard in the Maestro GUI [33].

For sampling the conformational space of the solvated dimer **12** ligand, the replica exchange solute tempering molecular dynamics (REST MD) method was applied with 3 replicas in DMSO and water solution using 300K as the lowest energy replica [33].

Firstly, the ligand–protein complexes were simulated by docking studies using the Glide package of the Schrodinger suite [34]. An extra-precision (XP) docking protocol was selected with enhanced ligand sampling, and the taxoid-binding site was selected as a ligand-binding pocket. The best pose according to the Glide Emodel scoring function was selected as a possible binding geometry of the ligand and REST simulation was carried out to examine the stability of the complex. The docking calculations never showed such a binding pose where the open D-ring steroid part was fitted in the taxoid binding pocket. Including this option into our studies, we also built a starting complex where an open D-ring skeleton was fitted manually to the binding pocket and REST studies were performed with this ligand orientation as well. An OPLS4 forcefield with SPC explicit water model was applied in all 200 ns long REST calculations at physiological salt concentrations using the Desmond program of the Schrodinger suite [35,36]. Finally, the binding free energy was calculated by the molecular mechanical level generalized Born surface area continuum solvation method (MM/GBSA) [37].

## 5. Conclusions

Steroidal heterodimers were designed based on the click conjugation of the earlier identified potent core-modified estrone derivatives. The CuAAC reactions of the two antiproliferative steroidal azide or alkyne building blocks resulted in a potent heterodimer **12**. The most relevant mechanistic hallmarks of the anticancer action of **12**, i.e., the increase in the G2/M cell population at the expense of the G1 phase and the induction of apoptotic cell death, are shared by its structural elements. However, our dimer-constructing strategy conferred some pharmacological improvements compared with those of the building blocks. Azidoalcohols with an intact D-ring were generally less potent, especially against breast cancer cell lines. The cell-growth-inhibitory potential of dimer **12** is more pronounced on four cancer cell lines than those of the previously described 17-hydroxy-16-(triazolylphenyl) derivatives. It should be emphasized that the preliminarily established structure–activity relationship, namely, the higher antiproliferative potential of 17α-hydroxy-16β-triazolylphenyl diastereomers, is retained in dimers. The D-secoestrone part, on the other hand, was similarly potent against some cells, including HeLa and C33A, but was substantially less active on further gynecological cell lines.

As the computational simulations show, both core-modified steroidal parts might fit well in the tubulin protein’s taxoid binding pocket. This result draws attention to the flexible nature of the taxoid binding site, which should be considered in other computational studies. Although the test compound has alcohol and triazole moieties capable of forming strong hydrogen bonds, hydrophobic interactions and water bridges dominate.

Our results support a previously established fact concerning the altered properties of steroidal hybrid compounds. Even though the molar mass is doubled, the dimer may have more promising biological properties than its building blocks. It should be pointed out that dimer **12** shows a broader antiproliferative spectrum than its building blocks. The substantial action of compound **12** against the frequently occurring HPV-16 positive cervical cancer cell line SiHa and the triple negative breast cancer cell line MDA-MB-231 are newly emerging effects that, in themselves, justify the implementation of dimerization. The latter strategy is further contributed by the dimer’s better tumor selectivity index on the non-malignant NIH/3T3 cell line than that of the clinically used cisplatin.

## Data Availability

Data are contained within the article and Appendix A.

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
