# Peer review of "Synthesis of Estrone Heterodimers and Evaluation of Their In Vitro Antiproliferative Activity"

_ijms, 2024, doi:10.3390/ijms25084274_

Round 1

Reviewer 1 Report

Comments and Suggestions for Authors

Bozsity et al. submitted a manuscript showing the anticancer potential of estrone heterodimers in an in vitro model. 

I have major comments regarding the manuscript:

  1. The abstract should be rewritten and smoothly shortly present the background, objectives, methodology, results, and conclusions. It is a mixture of unrelated sentences in its present form.
  2. Please, include at least two nonmalignant cell lines to examine the cytotoxic selectivity regarding anticancer research.
  3. Figure 3 doesn't correspond with Table 1 data. For example, among tested cell lines, SIHA appeared as the most sensitive with IC50 value = 0,97 uM. However, when apoptosis/necrosis were examined, no significant differences in % of apoptotic cells were observed up to 10 uM. Also, necrosis was not significantly different up to 3 uM in comparison to untreated cells. What was the explanation? 
  4. Cisplatin's effect on apoptosis/necrosis should also be examined as a reference to assess the molecular mechanisms of its cytotoxic action in comparison to tested dimers.
  5. For HeLa cells IC50= 1,7 Um whereas in Figure 3 after 10 uM treatment still more than 50% of cells were alive. It seems that the MTT assay was not the most suitable method for this analysis, because MTT and apoptosis results do not correspond with each other. 
  6. A more detailed analysis regarding molecular mechanisms of observed cytotoxicity should be applied or another tjan MTT assay test used.
  7. What was the solvent for steroidal heterodimers? The solvent control should be added to all analyses, not only non-treated control. 
  8. Sections 4.1.1.1-4.1.1.4 should be moved to the Supplementary data.
  9. Please correct References formatting and style, according to the IJMS template.

Comments on the Quality of English Language

Moderate editing of English language required

Author Response

Answers to Reviewer #1

We are grateful to the reviewer for his or her comments and here are our responses:

The abstract should be rewritten and smoothly shortly present the background, objectives, methodology, results, and conclusions. It is a mixture of unrelated sentences in its present form.

The abstract has been rewritten according to reviewer’s comments.

Please, include at least two non-malignant cell lines to examine the cytotoxic selectivity regarding anticancer research.

Authors have performed an additional MTT assay on NIH/3T3, a non-malignant mouse embryonic fibroblast, to expand the selectivity data of compound 12.

The results are from 2 independent experiments, n=5 each. We have inserted the results into Table 1 in the revised Manuscript.

inhibition, % ± SEM

conc.

11

12

13

14

cisplatin

cell lines

µM

Mean

SEM

Mean

SEM

IC50
(µM)

Mean

SEM

Mean

SEM

Mean

SEM

IC50
(µM)

NIH/3T3

10

n.d.

n.d.

41.39

2.76

15.02

n.d.

n.d.

n.d.

n.d.

76.74

1.26

4.73

30

n.d.

n.d.

68.64

3.74

n.d.

n.d.

n.d.

n.d.

96.90

0.25

Moreover, we have calculated selectivity indices for compound 12 and cisplatin; results can be found in Table 2 in the revised Manuscript. The interpretation of these recent results is also presented. The tumor-selectivity index was determined by dividing the mean IC50 against non-malignant cells by the mean IC50 against tumor cells, compared to MRC-5 and NIH/3T3 cells.

Tumor-selectivity index

NIH/3T3

MRC-5

cell lines

comp 12

cispl

comp 12

cispl

HeLa

8.78

0.38

0.73

0.36

SiHa

15.55

0.61

1.28

0.58

C33A

5.02

1.28

0.41

1.22

A2780

4.37

3.64

0.36

3.47

MCF-7

8.35

0.82

0.69

0.78

T-47D

14.47

0.48

1.20

0.46

MDA-MB-231

7.38

0.25

0.61

0.24

MDA-MB-361

7.26

1.28

0.60

1.22

MRC-5

--

--

1.00

1.00

NIH/3T3

1.00

1.00

--

--

Figure 3 doesn't correspond with Table 1 data. For example, among tested cell lines, SIHA appeared as the most sensitive with IC50 value = 0.97 uM. However, when apoptosis/necrosis were examined, no significant differences in % of apoptotic cells were observed up to 10 uM. Also, necrosis was not significantly different up to 3 uM in comparison to untreated cells. What was the explanation? 

To determine the IC50 values, a standard antiproliferative MTT assay was used, with 72 hours of incubation. 72 hours of incubation allows cells to grow; therefore, we can have information about the antiproliferative effect, while in shorter incubation time, MTT refers only to the direct cytotoxic effect.

Our HOPI and cell cycle data support our hypothesis that comp 12 needs more time to reach the antiproliferative ability. Therefore, for mechanistic study, we should use a shorter incubation period to discover the onset of the compound effect. An elevated concentration is usually needed in parallel with the shorter incubation period.

For the Reviewer's point that Siha cells were the most sensitive after 72 hours, while it shows a modest effect in HoPI and cell cycle results, our explanation is that in contrast to the other two cell lines, SiHa cells show remarkably elevated necrotic cells proportion after 24 hours incubation, instead of apoptosis. Apoptosis is an active, programmed, well-regulated, and controlled process; therefore, it requires time and can be further investigated over time, while necrosis is an accidental, sudden cell death, where the cells lose their integrity rapidly, and uncontrolled release of cellular contents occur, leading to a more toxic effect (Fink et al, 2005). Therefore, the elevated necrotic process instead of apoptosis in SiHa cells leads to a more toxic effect after 72 hours, which can be seen at MTT results.

Fink SL, Cookson BT. Apoptosis, pyroptosis, and necrosis: mechanistic description of dead and dying eukaryotic cells. Infect Immun. 2005 Apr;73(4):1907-16. doi: 10.1128/IAI.73.4.1907-1916.2005. PMID: 15784530; PMCID: PMC1087413.

Cisplatin's effect on apoptosis/necrosis should also be examined as a reference to assess the molecular mechanisms of its cytotoxic action in comparison to tested dimers.

Cisplatin was used as a reference compound during the antiproliferative screening because it is a routinely used drug in oncotherapy, especially in gynecological cancer cases. For further mechanistic studies, using cisplatin as a reference was not considered because our compounds have no structural similarities. Therefore, we hypothesize that we cannot estimate or compare any information about cisplatin results from our mechanistic studies. All our mechanistic experiments were performed using a self-control design based on these considerations.

For HeLa cells IC50= 1.7 Um whereas in Figure 3 after 10 uM treatment still more than 50% of cells were alive. It seems that the MTT assay was not the most suitable method for this analysis, because MTT and apoptosis results do not correspond with each other.

 and

A more detailed analysis regarding molecular mechanisms of observed cytotoxicity should be applied or another than MTT assay test used.

MTT assay is a standard, commonly used quantitative colorimetric assay for mammalian cell survival and proliferation. It is a widely utilized method to investigate the antiproliferative or cytotoxic effect of test compounds. Like many labs worldwide, our lab uses and publishes 72-hour MTT results during screening and IC50 determination of new compounds.

MTT assay does not refer to apoptosis results because it is a viability assay. This assay detects only living cells because mitochondria of viable cells generate the signal, so we have information about the cell survival compared to the controls. However, we cannot conclude any information on the cell death process from MTT. It seems appropriate to define the IC50 value, which is the concentration of compound required to inhibit cell growth by 50%. The IC50 value is crucial data for detailed investigations because if we decrease the incubation time, we need to elevate the drug concentration.

For the detailed mechanistic studies, we need more targeted methods, and we aim to find the appropriate incubation time and concentrations to discover the onset of action. That is why the 72 hours IC50 values from Table 1 are not the same as the concentration of the detailed investigations after 24 and 48 hours.

What was the solvent for steroidal heterodimers? The solvent control should be added to all analyses, not only non-treated control. 

Dimethylsulfoxide (DMSO) was the solvent for the investigated compounds.

10 mM stock solutions were prepared in DMSO from the 4 newly synthesized compounds. All the other dilutions were prepared from that stock solution in Eagle's Minimum Essential Medium (EMEM) with standard supplementation. Based on that process, the highest concentration, 30 µM of test compounds (15 µl stock solution in 5 ml EMEM), has only 0.3% (V/V) of DMSO. We have DMSO results on all of our cell lines, and it has no effect for the cells in 0.3% (all of the cell lines the growth inhibition of 0.3% DMSO solution in less than 3%); we can measure more than 10% growth inhibition just after 3% DMSO content, which is higher with on over of magnitude than the used concentration range.

Sections 4.1.1.1-4.1.1.4 should be moved to the Supplementary data.

Sections 4.1.1.1-4.1.1.4 have been moved to the Supplementary data. A sentence has been added to the Experimental section: “Characterization data of the reported compounds and NMR spectra are described in the Supporting Information.”

Please correct References formatting and style, according to the IJMS template.

The formatting and style were corrected.

Reviewer 2 Report

Comments and Suggestions for Authors

The paper presents the synthesis and antiproliferative activity of estrone heterodimers 11-14.

The chemistry strategy is simple and the reaction yields are good. The bioassay results for compound 12 are interesting.

In my opinion, the paper is worth studying and the manuscript contains enough original and interesting material to enrich the research so far. It requires some modifications before being published which I have outlined in the comments below:

The authors should emphasize the novelty and importance of the research conducted.

Selectivity indices for compound 12 and cisplatin should be given in a table and described.

Text formatting should be carefully checked.

References should be adapted to the journal's requirements.

Author Response

Answers to Reviewer #2

We are grateful to the reviewer for his or her comments and here are our responses:

1, The authors should emphasize the novelty and importance of the research conducted.

The novelty and importance of the research have been emphasized in the conlusions. This section was rewritten.

2, Selectivity indices for compound 12 and cisplatin should be given in a table and described.

Tumor-selectivity indices were determined by dividing the mean IC50 against non-malignant cells by the mean IC50 against tumor cells compared to both NIH/3T3 (mouse fibroblast) and MRC-5 (human fibroblast) cells.

 Briefly, tumor selectivity index results revealed that in the case of non-malignant NIH/3T3 cells, the selectivity of our compound 12 is better on all the investigated tumor cell lines compared with the clinically used cisplatin. In the case of non-malignant  MRC-5 cells, the results show comparable selectivity with cisplatin; the range of the selectivity indices is similar (comp 12: 0.36-1.28 and cispl: 0.24-3.47); in the case of HeLa, SiHa, T47D, and MDA-MB231 comp 12, while in the case of C33A, A2780, MDA-MB-361 cisplatin shows better selectivity and MCF-7 cells has almost the same selectivity indices.

Tumor-selectivity index

NIH/3T3

MRC-5

cell lines

comp 12

cispl

comp 12

cispl

HeLa

8.78

0.38

0.73

0.36

SiHa

15.55

0.61

1.28

0.58

C33A

5.02

1.28

0.41

1.22

A2780

4.37

3.64

0.36

3.47

MCF-7

8.35

0.82

0.69

0.78

T-47D

14.47

0.48

1.20

0.46

MDA-MB-231

7.38

0.25

0.61

0.24

MDA-MB-361

7.26

1.28

0.60

1.22

MRC-5

--

--

1.00

1.00

NIH/3T3

1.00

1.00

--

--

3, Text formatting should be carefully checked.

The text formatting has been checked.

4, References should be adapted to the journal's requirements.

The references have been corrected.

Round 2

Reviewer 1 Report

Comments and Suggestions for Authors

The authors significantly improved the manuscript and addressed all my questions. Thank you. I recommend its acceptance.

Comments on the Quality of English Language

 Minor editing of English language required